# Extra Proximal-Gradient Network with Learned Regularization for Image Compressive Sensing Reconstruction

**DOI:** 10.3390/jimaging8070178

**Published:** 2022-06-23

**Authors:** Qingchao Zhang, Xiaojing Ye, Yunmei Chen

**Affiliations:** 1Department of Mathematics, University of Florida, Gainesville, FL 32611, USA; qingchaozhang@ufl.edu; 2Department of Mathematics and Statistics, Georgia State University, Atlanta, GA 30303, USA; xye@gsu.edu

**Keywords:** image reconstruction, deep learning, learned optimization algorithm

## Abstract

Learned optimization algorithms are promising approaches to inverse problems by leveraging advanced numerical optimization schemes and deep neural network techniques in machine learning. In this paper, we propose a novel deep neural network architecture imitating an extra proximal gradient algorithm to solve a general class of inverse problems with a focus on applications in image reconstruction. The proposed network features learned regularization that incorporates adaptive sparsification mappings, robust shrinkage selections, and nonlocal operators to improve solution quality. Numerical results demonstrate the improved efficiency and accuracy of the proposed network over several state-of-the-art methods on a variety of test problems.

## 1. Introduction

Recent years have witnessed the substantial success of deep neural networks (DNN) in a large variety of real-world applications [1,2,3,4,5,6,7,8,9]. Equipped with proven expressive power, DNNs can be used to approximate highly complicated functions provided a sufficient amount of data [10]. However, training DNNs as end-to-end black-boxes can be extremely data demanding, rendering DNNs difficult to interpret, generalize, and sensitive to noise and outliers. To overcome these issues, learned optimization algorithms (LOAs) have started to gain attention as they are designed to combine the interpretable mechanism of optimization algorithms and the expressive power of DNNs. One of the most important applications of LOAs is solving the inverse problem of general form
(1)$$\min_{x}\;f\left(x;y\right)+g\left(x\right),$$
where *f* is the data fidelity term determined by the data formation and noise distribution that relate the target solution x and the given measurement data y, and *g* is the critical (possibly nonconvex) regularization term that promotes the desired solution x, as *f* is often underdetermined and the data y can be incomplete and noisy. In classical approaches to inverse problems, the regularization *g* is often handcrafted based on human heuristics and limited experience, which can be overly simplified and not capable to capture the intrinsic complex features of the solution. LOAs, on the other hand, allow the regularization *g* to be learned from training data and hence can result in significant improvement over the handcrafted regularizations.

Our goal in this paper is to propose an efficient extra proximal gradient algorithm that employs the Nesterov’s acceleration technique and the extra gradient scheme, and unroll this algorithm into a deep neural network called the extra proximal gradient network (EPGN) to solve a class of inverse problems (Equation 1). Motivated by the least absolute shrinkage and selection operator (LASSO) [11,12,13], our EPGN implicitly adopts an $l_{1}$-type regularization in (Equation 1) with a nonlinear sparsification mapping learned from data. The proximal operator of this regularization is elaborated by several linear convolutions, nonlinear activation functions, and shrinkage operations for robust sparse feature selection in EPGN. As our focus application is in image reconstruction, we also incorporate a nonlocal feature selection component into the learned regularization to leverage similar patterns within images and improve reconstruction quality. The proposed EPGN combines the advantages of the accelerated extra gradient scheme, the sparsity promoting nonlinear transforms, and the nonlocal feature selections. As a consequence, our EPGN is efficient, robust, and accurate in a variety of image reconstruction problems as demonstrated by the numerical experiments.

## 2. Related Work

One of the early LOAs is the learned iterative shrinkage thresholding algorithm (LISTA) for solving $l_{1}$ regularized linear inversion [14]. LISTA maps the standard ISTA optimization algorithm to a recurrent neural network (RNN) with certain layer weights learned from training data to improve the performance. The asymptotic linear convergence rate for LISTA is established in [15,16]. Several variations of LISTA are proposed for image reconstruction with regularizations based on low rank or group sparsity [17], $l_{0}$ minimization [18], and learned approximate message passing [19]. These LOA methods employ handcrafted regularizations and require a closed-form solution of the proximal operator of the regularization term. The idea of LISTA is also extended to solve composite problems with linear constraints, called differentiable linearized alternating direction method of multipliers (D-LADMM) [20], which exhibits an asymptotic linear convergence rate.

To learn more general and adaptive regularization function in (Equation 1), the other group of LOAs is proposed to solve inverse problem (Equation 1) with learnable regularization. A straightforward approach in this group uses deep convolutional neural network (CNN), denoted by $h_{k}\left(\cdot \right)$, to replace the proximal operator ${\mathrm{prox}}_{\alpha_{k}g}$ [21] of the unknown regularization term *g* in the proximal gradient update:(2)$$\begin{array}{c} x_{k+1}={\mathrm{prox}}_{\alpha_{k}g}\left(b_{k}\right), \end{array}$$
where $\alpha_{k}>0$ is the step size in the *k*th iteration, $b_{k}:=x_{k}-\alpha_{k}\nabla f\left(x_{k};y\right)$, and ${\mathrm{prox}}_{g}\left(\cdot \right)$ is defined by
(3)$${\mathrm{prox}}_{g}\left(b\right):=\underset{x}{\arg\min}\;\frac{1}{2}{∥x-b∥}^{2}+g\left(x\right).$$

Therefore, one avoids explicit formation of the regularization *g*, but creates a neural network with prescribed *K* phases, where each phase mimics one iteration of the proximal gradient method such as (Equation 2) to compute $b_{k}$ as above and $x_{k}=h_{k}\left(b_{k}\right)$. The CNN $h_{k}$ can also be cast as a residual network (ResNet) [22] to represent the discrepancy between $b_{k}$ and the improved $x_{k}$ [23]. Such a paradigm is also embedded into half quadratic splitting [23], ADMM [24], and primal dual methods [25] to replace the proximal operator in the subproblems. To improve over the generic black-box CNNs above, several LOA methods are proposed to unroll numerical optimization algorithms such as deep neural networks so as to preserve their efficient structures with proven efficiency, such as the ADMM-Net [26] and ISTA-Net [27]. These methods also prescribe the phase number *K* and map each iteration of the corresponding numerical algorithm to one phase of the network, and learn specific components of the network using training data.

## 3. Extra Proximal Gradient Network

In this section, we propose a novel parameter-efficient deep neural network architecture to solve the inverse problem (Equation 1) with regularization learned from data. To this end, we first introduce the accelerated extra proximal gradient algorithm that combines Nesterov’s acceleration technique and the extra proximal gradient update in Section 3.1. In Section 3.2, we mimic this algorithm to construct the proposed extra proximal gradient network (EPGN), where Nesterov’s acceleration step corresponds to a simple linear combination layer in EPGN to boost convergence, and the extra proximal gradient structure induces a predictor–corrector update scheme with efficient utilization of network parameters in EPGN. For the image reconstruction applications considered in our experiment part, we integrate mixing layers into EPGN to combine the local and nonlocal image features for enhanced reconstruction quality in Section 3.3. Additional details of the EPGN training process are provided in Section 3.4.

### 3.1. Extra Proximal Gradient Algorithm

The extra gradient method proposed in the seminal work [28] has attracted significant interest in optimization in recent years. It has been extended to solve variational inequality problems [29] and convex/nonconvex composite optimization problems [30] with theoretical performance guaranteed. Extra gradient algorithms use an additional gradient step in a first-order optimization algorithm to improve the convergence results. This can also be interpreted as a predictor–corrector scheme to speed up convergence. The following two variants of the original extra gradient algorithm are closely related to our proposed extra proximal gradient algorithm. The first one is the extended extra gradient method in [30], which uses extra proximal gradient steps at each iteration to solve nonconvex composite minimization problem (Equation 1) by
(4a)$$\begin{array}{cc} x_{k+\frac{1}{2}} & ={\mathrm{prox}}_{\alpha_{k}g}\left(x_{k}-\alpha_{k}\nabla f\left(x_{k};y\right)\right), \end{array}$$
(4b)$$\begin{array}{cc} x_{k+1} & ={\mathrm{prox}}_{\beta_{k}g}\left(x_{k}-\beta_{k}\nabla f\left(x_{k+\frac{1}{2}};y\right)\right). \end{array}$$

The second one is the convex accelerated extra gradient algorithm developed in [31], which integrates Nesterov’s accelerated gradient method for smooth convex optimization [32] into the extra gradient scheme. Different from the classical extra gradient method, this algorithm evaluates gradients in both steps at an interpolation of the previous two iterates rather than the previous iterate only. Recall that Nesterov’s acceleration technique [32] for minimizing smooth convex function *f* is given by
(5a)$$\begin{array}{cc} {\tilde{x}}_{k} & =x_{k}+\gamma_{k}\left(x_{k}-x_{k-1}\right), \end{array}$$
(5b)$$\begin{array}{cc} x_{k+1} & ={\tilde{x}}_{k}-\alpha \nabla f\left({\tilde{x}}_{k};y\right), \end{array}$$
which performs a momentum structure (5a) to improve the convergence rate of standard gradient methods. For nonconvex problems, a monitor mechanism that tunes $\gamma_{k}$ adaptively can be introduced to remedy convergence issue [33]. Motivated by this acceleration technique, we propose to combine (4) and (5) and introduce the accelerated extra proximal gradient updating scheme summarized in Algorithm 1 to solve inverse problems of form (Equation 1). In Algorithm 1, $\alpha_{k}$ and $\beta_{k}$ are step sizes, and $\gamma_{k}$ is the momentum coefficient in the *k*th iteration.
**Algorithm 1:** Accelerated Extra Proximal Gradient Algorithm.**Input:** Data y and initialization $x_{0}=x_{-\frac{1}{2}}$.**Output:**$x=x_{K}$.For k=0,1,2,⋯,K−1, do
(6a)x˜k=xk+γk(xk−xk−12),(6b)bk+12=x˜k−αk∇f(x˜k;y),(6c)xk+12=proxαkg(bk+12),(6d)x^k=xk+12+γk(xk+12−xk),(6e)bk+1=x^k−βk∇f(x^k;y),(6f)xk+1=proxβkg(bk+1).

### 3.2. Extra Proximal Gradient Network (EPGN)

We now cast Algorithm 1 as an LOA by mapping its iterations to the phases of a deep neural network. To this end, we select a phase number *K* (value to be specified in our experiment), and construct a deep neural network with *K* phases where each phase performs the updates described in (6). More specifically, we retain the same updates (6a), (6b), (6d) and (6e) in the *k*th phase of the network. As a result, (6a) and (6d) are simple linear combination layers to integrate the momentum term for acceleration, and (6b) and (6e) are gradient updates for improved fitting to the data. The parameters $\alpha_{k},\beta_{k},\gamma_{k}$ are all to be learned for every phase *k* (we set $\alpha_{k}=\beta_{k}$ for simplicity). The remaining updates (6c) and (6f) are replaced by a robust implicit ResNet-type update (we will make it explicitly computable later):(7)$$\begin{array}{cc} x_{k+l} & =b_{k+l}+r_{k}\left(x_{k+l}\right), \end{array}$$
where l=1/2,1, and the residual mapping $r_{k}$ plays a critical role of regularization that improves the quality of output $x_{k+l}$ in each phase *k*. In this paper, we parameterize $r_{k}$ as a composition of two nonlinear mappings, denoted by $G_{k}$ and ${\tilde{G}}_{k}$, such that:(8)$$\begin{array}{cc} r_{k}\left(x_{k+l}\right) & ={\tilde{G}}_{k}\circ G_{k}\left(x_{k+l}\right). \end{array}$$

In the remainder of this subsection, we show the details of the CNN structures of these two nonlinear mappings $G_{k}$ and ${\tilde{G}}_{k}$ and how to make the implicit residual update (7) explicit by leveraging the robust shrinkage selection operator.

#### 3.2.1. Nonlinear Feature Extraction Operator $G_{k}$

We parametrize the nonlinear operator $G_{k}$ as a multilayer convolutional network of the following structure:(9)$$G_{k}\left(x\right)=B_{k}\sigma \left(A_{k}D_{k}x\right),$$
where $D_{k}$ and $A_{k}$ are two linear convolutional operations that generate and convolve the local features of the input x, σ is a nonlinear activation function set to the rectified linear unit (ReLU) (i.e., σ(x)=max(x,0)), and $B_{k}$ is another linear convolution that fuses the activated local features. All the linear mappings $A_{k}$, $B_{k}$, and $D_{k}$ are realized as 3×3 convolutions. Hence, the size of the receptive field (RF) [34] of $G_{k}$ is 7×7.

The purpose of $G_{k}$ is to extract the main features of its input, such that these features can be easily refined by a robust feature selector. To this end, we employ the soft shrinkage selection operator in LASSO, which is the proximal operator of the $l_{1}$ norm and proved to be effective in selecting sparse outstanding features and suppressing noises of its input. More specifically, we consider $G_{k}\left(b\right)$ as the features prepared to be further pruned by shrinkage (as b is obtained by direct gradient update (6b) and (6e) which may contain undesired artifacts, and hence the features $G_{k}\left(b\right)$ need further refinement), and $G_{k}\left(x\right)$ as the refined feature obtained by pruning $G_{k}\left(b\right)$ using shrinkage. In other words, we expect $G_{k}\left(x_{k+l}\right)$ to be
(10)$$S_{k}\left(G_{k}\left(b_{k+l}\right)\right)={\mathrm{prox}}_{\theta_{k}{∥\cdot ∥}_{1}}\left(G_{k}\left(b_{k+l}\right)\right),$$
where the shrinkage threshold $\theta_{k}>0$ is also to be learned with $A_{k}$, $B_{k}$, and $D_{k}$. Note that the component-wise shrinkage operator $S_{k}$ in (10) has a closed form solution as ${\left[S_{k}\left(z\right)\right]}_{i}=\max(|z_{i}|-\theta_{k},0)\cdot z_{i}/\left|z_{i}\right|$ for each component $z_{i}$ of z. To further increase our network capacity, we set the convolution $B_{k}$ in $G_{k}$ to contain $N_{f}$ kernels ($N_{f}$ is set to 32 by default), each of size 3×3 in our implementation, hence $G_{k}\left(x\right)$ has $N_{f}$ channels at each pixel of x. The shrinkage operator $S_{k}$ in (10) is applied channel-wise with varying $\theta_{k,j}$ where $j=1,\cdots ,N_{f}$. Hence, the learnable parameters of $G_{k}$ include one convolution $D_{k}$ with $N_{f}$ kernels of size 3×3 and convolutions $A_{k}$ and $B_{k}$ with $N_{f}$ kernels of size $3\times 3\times N_{f}$, and those of $S_{k}$ are the shrinkage thresholds $\theta_{k}=\left\{\theta_{k,j}:j\in \left[N_{f}\right]\right\}$.

#### 3.2.2. Nonlinear Residual Resembling Operator ${\tilde{G}}_{k}$

Based on (8), the purpose of the nonlinear operator $\tilde{G}$ is to resemble the residual term using the refined feature $G_{k}\left(x\right)$, we can interpret $G_{k}$ and ${\tilde{G}}_{k}$ respectively as encoder and decoder in a symmetric form. More specifically, we parametrize ${\tilde{G}}_{k}\left(x\right)$ as ${\tilde{D}}_{k}{\tilde{A}}_{k}\sigma \left({\tilde{B}}_{k}x\right)$, where ${\tilde{A}}_{k}$, ${\tilde{B}}_{k}$, and ${\tilde{D}}_{k}$ are all 3×3 convolutional operators, and ${\tilde{D}}_{k}$ compresses the $N_{f}$ channels back to 1 channel according to $D_{k}$ in implementation.

Combining the parametrized nonlinear operators $G_{k}$ and ${\tilde{G}}_{k}$ and the shrinkage operator $S_{k}$ into (8), we obtain an explicit update rule of (7) given by
(11)$$\begin{array}{c} x_{k+l}=b_{k+l}+{\tilde{G}}_{k}\circ S_{k}\circ G_{k}\left(b_{k+l}\right). \end{array}$$

This update rule is employed in (6c) and (6f) for l=1/2,1 respectively in the *k*th phase.

To summarize, our proposed extra proximal gradient network (EPGN) of a prescribed *K* phases is constructed by unrolling Algorithm 1, where each phase executes (6) but with (6c) and (6f) substituted by (11). The flowchart of the computation in EPGN is shown in Figure 1. The proposed EPGN not only inherits the advantages of Algorithm 1 but also employs learnable feature selection operations. Hence, EPGN combines the following properties: (i) the simple linear momentum layers (6a) and (6d) for improved convergence; (ii) the extra proximal gradient updates (6b), (6e), and (11) mimic the predictor–corrector scheme for parameter-efficient network structure; (iii) learnable feature extraction, selection, and residual resembling operators $\left(G_{k},{\tilde{G}}_{k},S_{k}\right)$ in (11) to effectively improve solution quality as the input data flows through EPGN.

### 3.3. EPGN with Nonlocal Operator (NL-EPGN)

Nonlocal methods have proven effective for image reconstruction problems, such as in variational methods [35] and nonvariational approaches such as the notable BM3D algorithm [36]. Nonlocal operators can significantly improve image quality as they use image patches located in different regions to exploit the inherent self-similarity of images. Recently, the success of the nonlocal methods has motivated the investigation of the architecture of DNNs that have the ability to capture long-distance dependencies of the image. The deep network architecture for gray-scale and color image denoising in [37] is inspired by the projected gradient algorithm for solving a common variational image restoration model with a learnable nonlocal regularization. The nonlocal neural network proposed in [38] can be viewed as a generalization of the classical nonlocal mean in [35] that computes the response at a position as a weighted average of the image intensities at all positions. The weights implicitly depend on the feature maps in the patches with the size determined by the receptive fields.

To exploit repeated features in images for enhanced reconstruction quality, we adopt the idea in [38]. However, unlike [38] which only relies on nonlocal features, our NL-EPGN fuses local and nonlocal features of images using a combination operator learned through training data. More specifically, we propose NL-EPGN to integrate a nonlocal operator $N_{k}$ into the residual operator in (11), so that the features refined by the shrinkage operation $S_{k}$ can be passed to $N_{k}$ to leverage nonlocal features in images:(12)$$\begin{array}{cc} x_{k+l} & =b_{k+l}+{\tilde{G}}_{k}\circ N_{k}\circ S_{k}\circ G_{k}\left(b_{k+l}\right). \end{array}$$

The operator $N_{k}$ contains two main components: a nonlocal block $M_{k}$ that extracts nonlocal features of the input, and a nonlinear layer that combines the local and nonlocal features. The details of these two components are given as follows.

#### 3.3.1. Nonlocal Feature Extraction Block $M_{k}$

Our design of the nonlocal feature extraction block $M_{k}\left(x_{k}\right)$ follows the work [38] which computes a weighted average of features at all locations in an image. More precisely, let ${\left[z\right]}_{j}$ denote the input feature vector at position *j* and ${\left[v\right]}_{i}$ the response vector at position *i* of an image, then the nonlocal block $M_{k}$ computes ${\left[v\right]}_{i}$ by:(13)$${\left[v\right]}_{i}=\sum_{j}w_{ij}{\left[\varphi \left(z\right)\right]}_{j},$$
where the function φ computes a representation of the input signal at position *j*, and $w_{ij}$ is the normalized weight depending on the similarity between ${\left[z\right]}_{i}$ and ${\left[z\right]}_{j}$. The mapping φ corresponds to a learnable matrix $W^{\varphi }$ (implemented as 1×1 convolution). The weights are computed by embedded Gaussian:(14)$$w_{ij}=\frac{\exp({\left[W^{\alpha }z\right]}_{i}^{⊤}{\left[W^{\beta }z\right]}_{j})}{\sum_{j}\exp\left({\left[W^{\alpha }z\right]}_{i}^{⊤}{\left[W^{\beta }z\right]}_{j}\right)},$$
where both $W^{\alpha }$ and $W^{\beta }$ are implemented as $N_{f}/2$ convolutional filters of kernel size 1×1. We employ the bottleneck structure to reduce computation [38]. Hence, the nonlocal block $M_{k}$ in phase *k* is implemented as $v=M_{k}\left(z\right)$ where
(15)$$M_{k}\left(z\right)=\mathrm{softmax}\left({\left[W_{k}^{\alpha }z\right]}^{⊤}\left[W_{k}^{\beta }z\right]\right)W_{k}^{\varphi }\left(z\right).$$

#### 3.3.2. Local and Nonlocal Combination Layer

We propose to use a learnable combination layer of form $\sigma (C_{k}\left[z,v\right])$ to merge the input local feature z and nonlocal feature v obtained by nonlocal block $M_{k}$ in (15). That is, the nonlocal operator $N_{k}$ is defined by
(16)$$N_{k}\left(z\right)=\sigma \left(C_{k}\left[z,M_{k}\left(z\right)\right]\right).$$

In the *k*th phase, the inputs of $N_{k}$ are $z=S_{k}\circ G_{k}\left(b_{k+l}\right)$ for l=1/2,1 as shown in (12), [·,·] stands for the concatenation operator at each pixel, and $C_{k}$ corresponds to a set of learnable weight vectors which project the concatenated vector to a scalar at each pixel (implemented as 1×1 convolution). The flowchart of the nonlocal operator is shown in Figure 2.

### 3.4. Network Training

As discussed above, the proposed EPGN (or NL-EPGN) consists of *K* phases, where each phase imitates one iteration in the accelerated extra gradient Algorithm 1 with proximal steps (6b) and (6e) replaced by (11) (or (12) for NL-EPGN). The flowchart of variables in the *k*th phase is shown in Figure 3. The parameters in the *k*th phase are collectively denoted by $\Theta_{k}$, which includes the feature extraction operator $G_{k}=\left[A_{k},B_{k},D_{k}\right]$, the residual resembling operator ${\tilde{G}}_{k}=\left[{\tilde{A}}_{k},{\tilde{B}}_{k},{\tilde{D}}_{k}\right]$, the nonlocal operator $\left[W_{k}^{\alpha },W_{k}^{\beta },W_{k}^{\varphi },C_{k}\right]$, the momentum coefficient $\gamma_{k}$, and the shrinkage thresholds $\theta_{k}$. Let $\Theta =\{\Theta_{k}:0\leq k\leq K-1\}$ be the set of all network parameters. Then, given *N* training data pairs of form $\left\{\left(x^{\left(i\right)},y^{\left(i\right)}\right):1\leq i\leq N\right\}$, where $y^{\left(i\right)}$ is the input measurement data and $x^{\left(i\right)}$ is the corresponding ground truth of the *i*th pair, we define the loss function of Θ as
(17)$$L\left(\Theta \right)=\frac{1}{N}\sum_{i=1}^{N}{∥x_{K}\left(y^{\left(i\right)};\Theta \right)-x^{\left(i\right)}∥}_{2}^{2},$$
where $x_{K}\left(y;\Theta \right)$ denotes the output of the EPGN (i.e., the output of the last, *K*th phase) parametrized by Θ given input data y. The optimal network parameter $\Theta^{*}$ is obtained by minimizing the loss function (17) in the training process. After training, the EPGN with parameter $\Theta^{*}$ serves as a feed-forward neural network that can reconstruct high-quality image x given new measurement data y on the fly.

## 4. Numerical Experiments

In this section, we evaluate the performance of the proposed EPGN and NL-EPGN on several inverse problems in imaging reconstruction applications. We focus on the reconstruction problem in compressive sensing in our experiments, however, the proposed method can be easily adapted to other image reconstruction problems by changing the data-fidelity term accordingly. All the experiments are implemented, trained, and tested in the TensorFlow framework [39] on a desktop with an Nvidia GTX-1080Ti GPU and 11 GB of graphics card memory (NVIDIA Corporation, Santa Clara, CA, USA).In all tests, the network parameters Θ of EPGN/NL-EPGN are initialized using the Xavier method [40] and trained with the Adam optimizer [41] with learning rate $1\times 10^{-4}$ for 200 epochs. To evaluate the reconstruction quality, we use the average peak signal-to-noise ratio (PSNR).

### 4.1. Nature Images Compressive Sensing

We first test EPGN on the compressive sensing (CS) image reconstruction problem. In our experiment we use the *91 Images* dataset for training and *Set11* for testing [42]. For a fair comparison, we follow the same data preparation and result evaluation procedures in [27]. The ground truth data $\left\{x^{\left(i\right)}:1\leq i\leq N\right\}$ contains N=88,912 image patches with luminance components that are all randomly cropped into size 33×33 from *91 Images* dataset. We then generate a matrix with random Gaussian entries of size 10%n and 25%n, where $n=33^{2}$, and orthogonalize the rows. Then the measurement data for training is $\left\{y^{\left(i\right)}=\Psi x^{\left(i\right)}:1\leq i\leq N\right\}$. The testing data *Set11* preparation follows the same procedure as training data.

#### 4.1.1. Comparison with Existing Methods

We set the phase number K=9 for EPGN and K=7 for NL-EPGN in this test (as shown in Figure 4 where the PSNRs of the networks become saturated). Table 1 shows the comparison of the average PSNRs of the images reconstructed by EPGN/NL-EPGN versus several state-of-the-art image reconstruction methods, namely TVAL3 [43], D-AMP [44], IRCNN [23], ReconNet [42], DR^2^-Net [45], ISTA-Net^+^ [27], and DPA-Net [46], where the first two are classical optimization-based methods, and the last five are deep learning-based methods. The PSNR results of the first four methods and ISTA-Net^+^ in Table 1 are quoted from [27]. We observe that EPGN and NL-EPGN outperform all aforementioned algorithms, whereas NL-EPGN obtains the highest accuracy.

#### 4.1.2. Reconstruction Quality Assessment

Compared to the state-of-the-art ISTA-Net^+^ [27], both EPGN and NL-EPGN obtain better reconstruction results with a similar number of parameters as shown in Table 2. In particular, Figure 5 shows the reconstructed butterfly image with a CS ratio of 10%, from which we can see that the 9-phase EPGN and 7-phase NL-EPGN can both capture the inconspicuous detail of the butterfly wings at the lower left part in the zoomed-in images. Similarly, Figure 6 shows the reconstructed cameraman image with a CS ratio of 25%, where the 9-phase EPGN and 7-phase NL-EPGN have fewer artifacts in the background compared to ISTA-Net^+^, as observed in the lower right area of the zoomed-in images. Figure 7 presents the reconstruction results of the Barbara image in *Set11* from ISTA-Net^+^,the 9-phase EPGN, and the 7-phase NL-EPGN with a CS ratio of 10%. We can observe that the texture pattern of the scarf is better preserved by NL-EPGN due to the nonlocal operator.

#### 4.1.3. Parameter Efficiency

The number of network parameters in each phase of ISTA-Net${}^{+}$ is 37,442 [27]. The number of trainable parameters of each phase in EPGN is $\left\{G_{k}+{\tilde{G}}_{k}+\gamma_{k}+\alpha_{k}+\theta_{k}=32\times 3\times 3\times \left(1+32\times 2\right)+32\times 3\times 3\times \left(32\times 2+1\right)+1+2+32=37,475\right\}$. Similarly, the number of learnable parameters of each phase in EPGN is 41,571. Therefore, the number of network parameters in each phase of ISTA-Net${}^{+}$, EPGN, and NL-EPGN are very similar (NL-EPGN is about 10.9% more than ISTA-Net${}^{+}$ and EPGN). Figure 4 shows the reconstruction PSNR of these three methods versus phase number, from which we observe that NL-EPGN becomes saturated with phase number K≥7, whereas EPGN with K≥9 and ISTA-Net+ with K≥11. Nevertheless, as shown in Table 2, a 7-phase NL-EPGN has fewer network parameters than a 9-phase EPGN but achieves even higher PSNR.

We compare the reconstruction results of EPGN and ISTA-Net${}^{+}$ in a range of different phase numbers with a CS ratio of 25%, as shown in Figure 4. We observe that the PSNR values improve as the phase number increases and become saturated after K≥9. EPGN achieves a 0.3 dB higher PSNR on average than ISTA-Net${}^{+}$. To further demonstrate the superiority of the extra proximal-gradient method over extending network depth, we compare the 9-phase EPGN with the 15-phase ISTA-Net${}^{+}$, as shown in Table 2. Compared to the 15-phase ISTA-Net${}^{+}$ which extends the depth of the network by simply adding more phases, the 9-phase EPGN achieves better accuracy (0.27 dB higher) using much fewer parameters and similar reconstruction time. We compare the reconstruction performance of EPGN and NL-EPGN with CS ratios of 10% and 25%, the 7-phase NL-EPGN outperforms the 9-phase EPGN by 0.21 dB and 0.15 dB respectively, as shown in Table 1. We also compare the reconstruction results of NL-EPGN and EPGN in a range of different phase numbers with a CS ratio of 25%. The results are shown in Figure 4. We observe that NL-EPGN achieves an average of 0.2 dB PSNR better than EPGN. It is interesting that the PSNR of NL-EPGN shows no significant improvement after K=7. As shown in Table 2, the reconstruction time of the 7-phase NL-EPGN is approximate 7 to 8 times that of the 9-phase EPGN due to the time complexity of the nonlocal operator. However, the effect of the nonlocal operator is remarkable, NL-EPGN with 7 phases has a 0.15 dB PSNR improvement with 13.7% fewer parameters compared to EPGN with 9 phases. In Figure 8, we show the PSNR versus epoch using the proposed NL-EPGN and the state-of-the-art method ISTA-Net${}^{+}$ for image reconstruction with a CS ratio of 10% and phase number 3. While both networks gradually improve PSNR with more epochs, NL-EPGN appears to be significantly more effective than ISTA-Net${}^{+}$ during training as the former produces reconstructions with much higher PSNR.

### 4.2. MR Images Compressive Sensing

We also test the performance of EPGN on compressive sensing reconstruction of brain MR images [47] (CS-MRI). We randomly selected 100 and 50 images for training and testing, respectively, and cropped every image to the size of 190×190. In the CS-MRI problem, the data fidelity is $f\left(x;y\right)={∥\Phi x-y∥}_{2}^{2}$, where Φ=PF, *P* is a binary selection matrix representating the sampling trajectory, and F is the discrete Fourier transform. We compare EPGN with ISTA-Net${}^{+}$ [27] on the same MRI data set. The experimental results on various undersampling ratios of radial masks are summarized in Table 3. Here, we set the phase number of ISTA-Net${}^{+}$ and EPGN to 15 and 11 respectively. It is obvious that EPGN outperforms ISTA-Net${}^{+}$ for each undersampling ratio.

## 5. Concluding Remarks

We presented a novel deep neural network architecture, called the extra proximal gradient network (EPGN), to solve a general class of inverse problems with a focus on image reconstruction applications. EPGN imitates the accelerated extra proximal gradient algorithm and features a learned regularization that incorporates adaptive sparsification mappings, robust shrinkage selections, and the combination of local and nonlocal operators for improved solution quality and network parameter efficiency. Extensive numerical experiments show that EPGN outperforms several existing state-of-the-art methods on a variety of image reconstruction problems.

## Figures and Tables

**Figure 1 jimaging-08-00178-f001:**
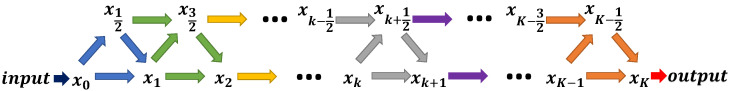
Overview of the *K*-phase extra proximal gradient network (EPGN) architecture. The arrows in the same color indicate computations within the same phase and share the same operators and parameters.

**Figure 2 jimaging-08-00178-f002:**
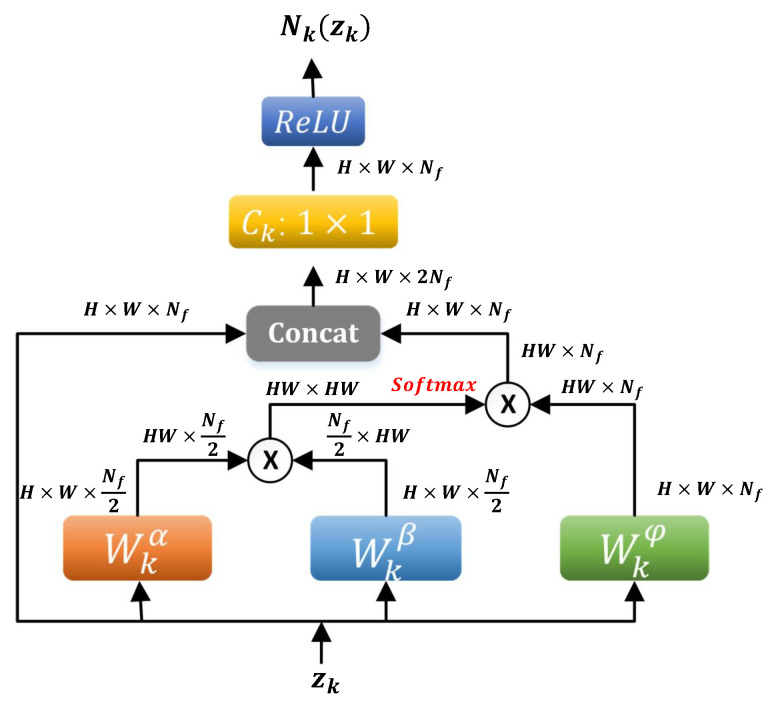
Data flow in the nonlocal operator $N_{k}$ (16). “⨂” denotes matrix multiplication. Both input $z_{k}$ and output $N(z_{k})$ are of the same shape $H\times W\times N_{f}$ (height × width × #channel) of the image.

**Figure 3 jimaging-08-00178-f003:**
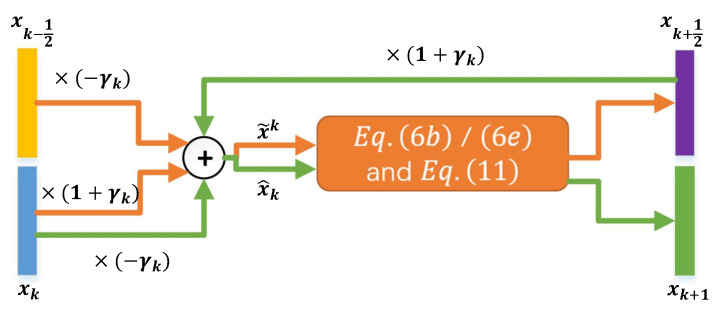
Data flow in the *k*th phase of EPGN. “⨁” represents element-wise sum. Orange and green arrows represent computations in the first ((6a), (6b) and (11) with l=1/2) and second ((6d), (6e) and (11) with l=1) stages of EPGN, respectively.

**Figure 4 jimaging-08-00178-f004:**
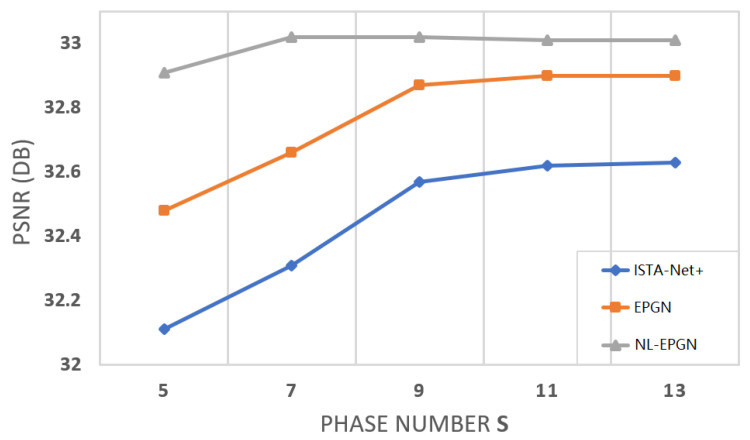
Average PSNR comparison between ISTA-Net${}^{+}$, EPGN and NL-EPGN with various phase number on image compressive sensing problem on *Set11* with a CS ratio of 25%.

**Figure 5 jimaging-08-00178-f005:**
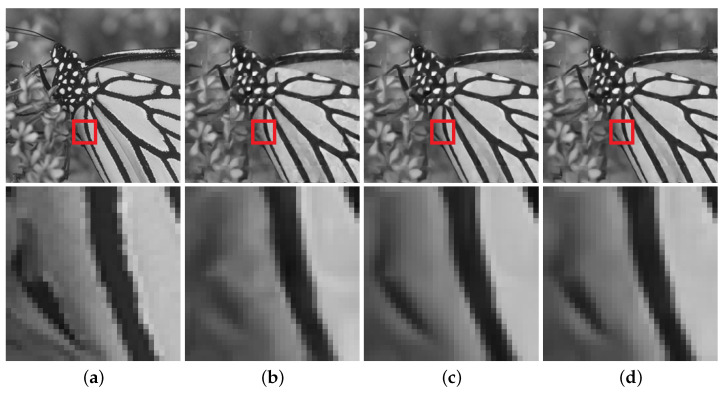
Reconstruction of a butterfly image with a CS ratio of 10% using the 9-phase ISTA-Net${}^{+}$ (PSNR 25.91dB), 9-phase EPGN (26.47dB), and 7-phase NL-EPGN (26.58dB). (**a**) Ture. (**b**) ISTA-Net${}^{+}$. (**c**) EPGN. (**d**) NL-EPGN.

**Figure 6 jimaging-08-00178-f006:**
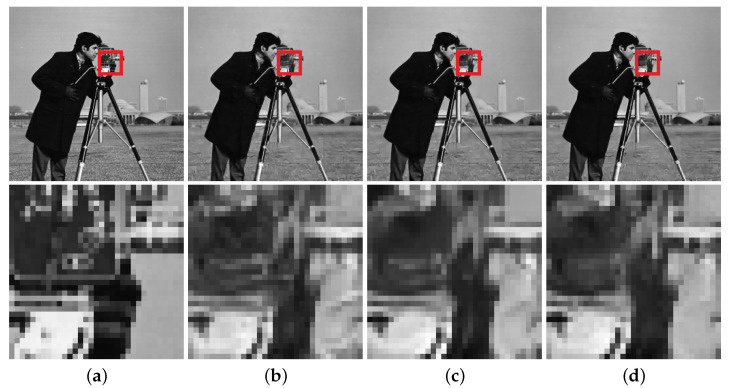
Reconstruction of the cameraman image with a CS ratio 25% using the 9-phase ISTA-Net${}^{+}$ (PSNR 28.97dB), 9-phase EPGN (29.62dB), and 7-phase NL-EPGN (29.73dB). (**a**) Ture. (**b**) ISTA-Net${}^{+}$. (**c**) EPGN. (**d**) NL-EPGN.

**Figure 7 jimaging-08-00178-f007:**
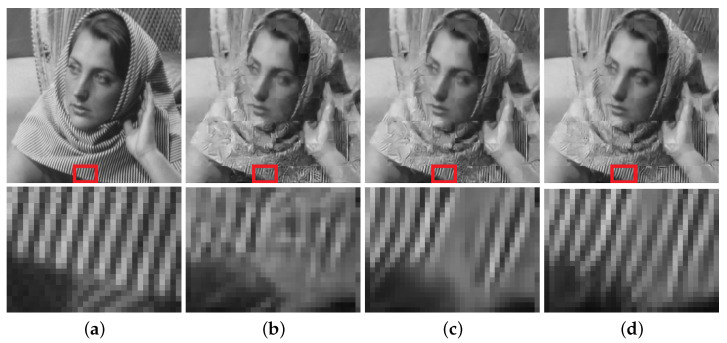
Reconstruction of the Barbara image with a CS ratio of 10% using the 9-phase ISTA-Net${}^{+}$ (PSNR 23.59dB), 9-phase EPGN (23.89dB), and 7-phase NL-EPGN (24.27dB). (**a**) Ture. (**b**) ISTA-Net${}^{+}$. (**c**) EPGN. (**d**) NL-EPGN.

**Figure 8 jimaging-08-00178-f008:**
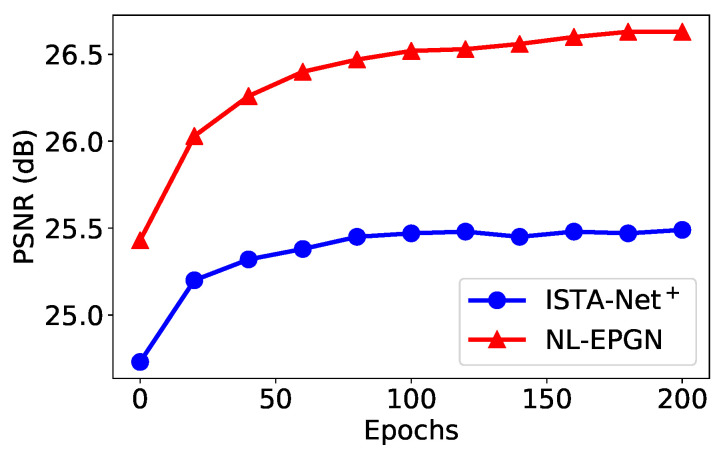
Average PSNR comparison between ISTA-Net${}^{+}$ and NL-EPGN with various numbers of epochs during training with 3 phases on *Set11* with a CS ratio of 10%.

**Table 1 jimaging-08-00178-t001:** Natural image CS reconstruction results by existing methods and the proposed EPGN (with 9 phases) and NL-EPGE (with 7 phases) on dataset *Set11* with CS ratios of 10% and 25%. Table shows the average PSNR (dB) of the comparison methods.

Method	10%		25%	
	PSNR	SSIM	PSNR	SSIM
TVAL3 [43]	22.99	0.3758	27.92	0.6238
D-AMP [44]	22.64	-	28.46	-
IRCNN [23]	24.02	-	30.07	-
ReconNet [42]	24.28	0.6406	25.60	0.7589
DR${}^{2}$-Net [45]	24.32	0.7175	28.66	0.8432
ISTA-Net^+^ [27]	26.64	0.8036	32.57	0.9237
DPA-Net [46]	26.99	0.8354	31.74	0.9238
EPGN (9-phase)	27.12	0.8893	32.87	0.9611
NL-EPGN (7-phase)	27.33	0.8956	33.02	0.9623

**Table 2 jimaging-08-00178-t002:** Compressive sensing reconstruction performance comparison of the 9-phase ISTA-Net${}^{+}$, 15-phase ISTA-Net${}^{+}$, 9-phase EPGN, and 7-phase NL-EPGN on *Set11* with a CS ratio of 25% on the number of network parameters (# PARM), average PSNR in dB with standard deviation over the reconstructed images, and average reconstruction time (Time) of one image in second.

Network (# Phase)	# PARM	PSNR (dB)	Time (s)
ISTA-Net${}^{+}$ (9)	336,978	32.57 ± 2.20	0.084
ISTA-Net${}^{+}$ (15)	561,630	32.60 ± 2.19	0.103
EPGN (9)	337,275	32.87 ± 2.24	0.110
NL-EPGN (7)	290,997	33.02 ± 2.05	0.802

**Table 3 jimaging-08-00178-t003:** PSNR (dB) of reconstructions obtained by ISTA-Net${}^{+}$ and EPGN on MR images using radial masks with sampling ratios of 10%, 20%, and 30%.

Method	10%	20%	30%
ISTA-Net${}^{+}$	33.49	40.66	44.70
EPGN	33.70	40.94	45.45

## Data Availability

The MRI data presented in this study are available on 2013 Diencephalon Free Challenge [47] at https://my.vanderbilt.edu/masi/workshops/ (accessed on 5 April 2022) and the nature image data are available at https://people.ee.ethz.ch/~timofter/ (accessed on 5 April 2022).

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
