# Peer review of "Extra Proximal-Gradient Network with Learned Regularization for Image Compressive Sensing Reconstruction"

_2313-433X, 2022, doi:10.3390/jimaging8070178_

Round 1
Reviewer 1 Report
The paper proposes two neural networks with learned regularization to solve inverse problems. The idea is fine and well presented. However in my opinion there are some concerns, mainly in numerical experiments.
1) the title is too general: the proposed networks are tested only for the solution of a single inverse problem, i.e. the reconstrcution from compressed sensing.
2) the authors say that the methods are tested on a variety of image reconstruction problems. Really, only the results on three very common images are reported. Moreover, neither noise nor blur Is added on the input data.
3) Only PSNR is used to evaluate the results. I think that other metrics can be introduced, such as, for example, SSIM and Mean Square Error.
4) The values of PSNR are reported only for the test images shown in the paper. I think that the authors should test their network on a test set and report the values of the metrics considered on the whole test set (possibly with variance or confidence intervals)
Reviewer 2 Report
The authors presented a deep neural network architecture, called the extra proximal gradient network. The method is clearly described and has been experimentally demonstrated to have high performance and parameter efficiency compared to some other well-known algorithms but the following comments could be incorporated to improve the manuscript.
(1) Explain that the number of epoch described in section 4 is fair for all the methods compared. For example, the authors can include a PSNR diagram compared to various numbers of epoch during training.
(2) Are there any restrictions on the research conducted?
Round 2
Reviewer 1 Report
The authors have modified the paper according to the suggestions.
Autocitations 24 and 36 are inapproèriate in my opinion and they should be eliminated.
Author Response
Thanks for the comments, we eliminated the autocitations 24 and 36 in the revised paper.